# Synthesis, Anti-Inflammatory, and Molecular Docking Studies of New Heterocyclic Derivatives Comprising Pyrazole, Pyridine, and/or Pyran Moieties

**DOI:** 10.3390/ph18030335

**Published:** 2025-02-26

**Authors:** Mohamed A. M. Abdel Reheim, Hend S. Abdel Rady, Omnia A. Mohamed, Abdelfattah Hassan, Ibrahim S. Abdel Hafiz, Hala M. Reffat, Fahmy Gad Elsaid, Mamdouh Eldesoqui, Dalal Sulaiman Alshaya, Abdelnaser A. Badawy, Eman Fayad, Aboubakr H. Abdelmonsef

**Affiliations:** 1Department of Chemistry, Faculty of Science, Arish University, Arish 45511, Egypt; m_abdelreheim@aru.edu.eg (M.A.M.A.R.); hend.saad@sci.aru.edu.eg (H.S.A.R.); ibrahim.saad@sci.aru.edu.eg (I.S.A.H.); halarefat@aru.edu.eg (H.M.R.); 2Department of Biochemistry and Molecular Biology, Theodor Bilharz Research Institute, Giza 12411, Egypt; oali19871987@gmail.com; 3Department of Medicinal Chemistry, Faculty of Pharmacy, South Valley University, Qena 83523, Egypt; abdelfattah_hassan@svu.edu.eg; 4Department of Biology, College of Science, King Khalid University, P.O. Box 960, Abha 61421, Saudi Arabia; felsaid@kku.edu.sa; 5Department of Basic Medical Sciences, College of Medicine, AlMaarefa University, P.O. Box 71666, Riyadh 11597, Saudi Arabia; mamrah@um.edu.sa; 6Department of Biology, College of Science, Princess Nourah bint Abdulrahman University, P.O. Box 84428, Riyadh 11671, Saudi Arabia; dsalshaya@pnu.edu.sa; 7Department of Biochemistry, Faculty of Medicine, Northern Border University, P.O. Box 1321, Arar 91431, Saudi Arabia; abdelnser.ali@nbu.edu.sa; 8Department of Biotechnology, College of Sciences, Taif University, P.O. Box 11099, Taif 21944, Saudi Arabia; hhhh_fayed@yahoo.co.uk; 9Department of Chemistry, Faculty of Science, South Valley University, Qena 83523, Egypt

**Keywords:** anti-inflammatory, Claisen–Schmidt reaction, Michael addition, molecular docking, pyran, pyrazole, pyridine

## Abstract

**Introduction:** Inhibiting cyclooxygenase-2 (COX-2) is a potential strategy in inflammation therapy. Thus, developing COX-2 inhibitors plays a pivotal role in efficient inflammation treatment. This study discloses the synthesis of new heterocyclic compounds incorporating pyridine, pyran, and/or pyrazole moieties as COX-2 inhibitors. **Methods**: In this study, the Claisen–Schmidt reaction of 1-(5-hydroxy-1,3-diphenyl-1*H*-pyrazol-4-yl)ethan-1-one **1** and *p*-methoxybenzaldehyde in ethanol containing aqueous sodium hydroxide (10%) led to the formation of 1-(5-hydroxy-1,3-diphenyl-1*H*-pyrazol-4-yl)-3-(4-methoxyphenyl)prop-2-en-1-one) **2**. The latter compound was allowed to react as a key precursor with various nucleophiles such as ethyl cyanoacetate, malononitrile, cyclohexanone, ethyl acetoacetate, hydrazine, cyano acid hydrazide, hydrazide, and/or thiosemicarbazide to yield new heterocyclic derivatives comprising pyridine, pyran, and/or pyrazole moieties **3**–**15**, according to the Michael addition reaction. The newly synthesized compounds were depicted using spectroscopic techniques such as IR, ^1^H-NMR, ^13^C-NMR, and MS. Moreover, their anti-inflammatory efficiency was in vitro evaluated by means of protein denaturation inhibition and cell membrane protection assay. **Results:** The results of 2^−ΔΔct^ values of COX-2 expression for compounds 6, 11, 12, and 13 were 6.6, 2.9, 25.8, and 10.1, respectively. Therefore, compound 12, followed by 13, 11, and 6, showed potent anti-inflammatory properties by in vitro evaluation. Further, an in silico molecular docking study was performed on the best-docked compounds and reference drug (Diclofenac) to investigate their binding affinities against the active site of the target enzyme. The obtained results from the in silico study aligned with the biological evaluation. **Conclusions**: The studies open new doors for designing new heterocycles containing pyridine, pyran, and/or pyrazole moieties as potent anti-inflammatory agents.

## 1. Introduction

Protein denaturation leads to the loss of the protein’s biological functions, and this process has been implicated in the development of various inflammatory disorders, including rheumatoid arthritis, diabetes, and cancer [1]. The ability of a substance to prevent protein denaturation is crucial in mitigating such inflammatory conditions [2]. Additionally, proteinases, which are abundant in the lysosomal granules of neutrophils, can cause tissue damage during inflammatory responses [3]. Consequently, the development of proteinase inhibitors is closely linked to the efficacy of anti-inflammatory drugs. Moreover, during inflammation, the lysosomal membrane may undergo lysis, releasing enzymes that contribute to various disorders [4]. Non-steroidal anti-inflammatory drugs (NSAIDs) exert their effects by either inhibiting the release of lysosomal enzymes or stabilizing the lysosomal membranes [5]. The lysis of red blood cell (RBC) membranes, leading to hemolysis and hemoglobin oxidation, can expose RBCs to harmful substances, preventing the leakage of their intracellular components [6]. Subsequently, the inhibition of hypotonicity and heat-induced lysis of the RBC membrane is a valuable measure of anti-inflammatory activity. This approach is particularly relevant because the structure of human RBC membranes closely resembles that of lysosomal membranes. Hypotonic solutions can induce excessive fluid accumulation within RBCs, causing membrane rupture and subsequent hemolysis. Real-time PCR detection of the COX-2 gene offers a highly sensitive and specific approach for quantifying gene expression, especially in the study of inflammation and the evaluation of anti-inflammatory therapies [7,8,9,10]. COX-2, an enzyme critical for the synthesis of pro-inflammatory prostaglandins, is frequently upregulated during inflammatory responses [11,12]. Through real-time PCR, researchers can precisely quantify changes in COX-2 mRNA levels following the administration of anti-inflammatory agents, yielding crucial insights into the gene’s involvement in inflammation and the efficacy of therapeutic interventions [13]. This technique enables detailed monitoring of gene expression dynamics, enhancing our understanding of inflammatory mechanisms and informing the development of effective treatments.

Chalcone, also known as α,β-unsaturated ketone, is an excellent precursor for the production of various heterocyclic molecules [14,15]. Chalcone derivatives have a wide spectrum of biological activities, including anti-cancer [16,17], anti-inflammatory [18], anti-oxidant [19], and antimicrobial [20]. In addition, the literature suggested that they serve as potential candidates for modulating inflammatory pathways [21]. Further, various synthetic compounds containing pyrazole, pyridine, and/or pyran moieties demonstrate the most important heterocycles, widely possessing a broad spectrum of anti-inflammatory efficiencies [22,23,24], as declared in Figure 1.

In light of the aforementioned and as part of our ongoing efforts to synthesize bioactive molecules with significant biological properties, herein, we have designed and synthesized a new series of heterocyclic compounds incorporating pyridine, pyran, and/or pyrazole moieties as COX-2 inhibitors. Additionally, their anti-inflammatory efficiencies were in vitro examined using protein denaturation inhibition and cell membrane protection assays. Furthermore, the docking investigations were utilized to acquire a better understanding of the binding mode of the newly prepared compounds against the COX-2 target enzyme.

## 2. Results and Discussion

### 2.1. Chemistry

The reaction routes for the synthesis of the title compounds were outlined in Figure 1, Figure 2, Figure 3 and Figure 4. In the present work, the key precursor **2** was produced by the reaction of compound **1 [25]** with *p*-methoxybenzaldehyde, according to base-catalyzed Claisen–Schmidt condensation (Figure 1). Because of the ketoethylenic moiety (CO-CH=CH), chalcone has strong reactivity toward several nucleophilic reagents. The behavior of certain active methylene reagents toward α,β-unsaturated ketones was investigated. Thus, when chalcone **2** is allowed to react with ethyl cyanoacetate in refluxing sodium ethoxide solution, it affords the pyran derivative **3** (Figure 1). The confirmation of structure **3** was based on its elemental analysis and spectral data. Thus, the IR spectrum of compound **3** showed absorption bands at 3449 cm^−1^, which corresponded to elongation vibration of the OH bond, and 2218 cm^−1^, which corresponded to elongation vibration of the nitrile bond (CN). In the mass spectrum, the molecular ion peak at *m/z* (%) = 463 (M^+^) was found to correlate with its accurate molecular formula C_28_H_21_N_3_O_4_. Initially, chalcone **2** and ethyl cyanoacetate underwent an intramolecular cyclocondensation reaction in refluxing ethanol in the presence of ammonium acetate, producing compound **4** bearing a new pyridone moiety (Figure 1). The assigned structure is compatible with the elemental and spectroscopic data. IR spectrum exhibited absorption bands at 3448–3400, 2209, and 1707 cm^−1^ due to the presence of (OH/NH), CN, and C=O groups, respectively. Additionally, the ^1^H-NMR spectrum exhibited signals at ™ 3.70 (s, 3H, OCH_3_), 6.60–8.07 (m, 15H, Ar-H+ H-_pyridone_), 9.90 (hump, 1H, NH), and 11.00 (hump, 1H, OH). Finally, the mass spectrum supports the suggested structure, as it displayed an extremely strong molecular ion peak at *m/z* (%) = 460 (M^+^).

Similar to the behavior of compound **2** toward ethyl cyanoacetate, the short-term condensation of chalcone **2** and malononitrile in glacial acetic acid yielded 2-(3-(5-hydroxy-1,3-diphenyl)-1H-pyrazol-4-yl)-1-(4-methoxyphenyl)-3-oxopropyl)malononitrile **5**, which when refluxed in glacial acetic acid with sodium acetate, produced the equivalent amino dihydropyrancarbonitrile derivative **6**. Chalcone **2** reacted with malononitrile in an ethanolic sodium ethoxide solution to furnish the later product **6**, which was also directly yielded. The IR spectrum of pyran derivative **6** displayed absorption bands at 3448–3300 cm^−1^ corresponding to the -OH and -NH_2_ groups. Two singlet signals were observed in the ^1^H-NMR spectrum of compound **6** at δ 4.60 ppm and 6.04 ppm, referring to 4*H*-pyran and -NH_2_ groups, respectively. Along with a multiplet signal at δ 7.27–7.83 ppm for aromatic protons in an interference with the 5*H*-pyran proton. In accordance with its molecular formula, the mass spectrum also declared the molecular ion peak at *m/z* (%) = 463 (M^+^+1). In addition, the treatment of compound **2** with malononitrile in glacial acetic acid in the presence of ammonium acetate can be used as a step in the synthesis of the pyridine ring. Additionally, compound **7** was created by directly reacting 4-acetylpyrazole-5-ole **1** with substituted aromatic aldehydes and malononitrile in glacial acetic acid under reflux while also being in the presence of ammonium acetate [26] (Figure 2). The assigned structure of compound **7** is consistent with the results of elemental analyses and spectral data such as IR, ^1^H-NMR, and mass spectroscopy.

Literature surveys revealed that pyridines and fused pyridines are examples of heterocyclic compounds that are of a wide spectrum of biological actions, including those of anti-inflammatory and anti-tumor agents [27]. This prompted us to prepare a new fused pyridine system. Therefore, when chalcone **2** and cyclohexanone are allowed to react together in boiling glacial acetic acid containing ammonium acetate, a fused pyridine derivative **8** is produced (Figure 3). The spectroscopic techniques such as IR, ^1^H-NMR, and mass spectra were utilized to confirm the latter product’s structure. Similar to this, pyridone derivative **9** was prepared when chalcone **2** and ethyl acetoacetate were condensed in boiling glacial acetic acid containing ammonium acetate. Establishing structure **9** was based on elemental and spectral data.

Moreover, chalcone **2** was heated with ethyl acetoacetate in sodium ethoxide solution to yield the expected pyran derivative **10**. The chemical structure of compound **10** was elucidated by spectroscopic data. The IR spectrum exhibited stretching bands at 3402 cm^−1^ (OH) and 1720 cm^−1^ (C=O) groups. In the ^1^H-NMR spectrum, singlet signals were found at δ 2.00 ppm, which were assigned to the methyl protons, at δ 3.70 ppm to –OCH_3_ protons, at δ 5.23 ppm to 4H-pyran, a singlet signal at δ 6.01 ppm assignable to 5H-pyran, in addition to a multiplet signal for Ar-H at δ 6.87–7.83 ppm, a singlet signal at δ 10.38 ppm assignable to (OH), and a singlet signal at δ 11.80 ppm assigned to (OH). The proposed structure is supported by the mass spectrum; at *m/z* (%) = 480 (M^+^), a molecular ion peak was visible.

Our study focuses on the efficient utilization of α,β-unsaturated ketone in the ring closure reaction with binucleophilic reagents to synthesize substituted and fused pyrazole. Thus, chalcone **2** and hydrazine were cyclocondensed in refluxing ethanol to furnish the pyrazoline derivative **11**. On the other hand, the equivalent 1-acetylpyrazoline derivative **12** was obtained by cyclization of chalcone **2** with hydrazine in acetic acid. The pyrazoline derivative **11** also reacted with glacial acetic acid to give the product **12**. The spectral and analytical data support the proposed structure (see the experimental section).

Furthermore, the reaction of chalcone **2** with cyanoacetic acid hydrazide in boiling glacial acetic acid containing ammonium acetate led to the creation of 2-amino-6-(5-hydroxy-1,3-diphenyl-1H-pyrazol-4-yl)-4-(4-methoxyphenyl)nicotinohydrazide **13** in a satisfactory yield. The spectroscopic data, such as IR, ^1^H-NMR, and mass, were used to affirm the chemical structure of the product (experimental section). The IR spectrum exhibited absorption bands at 3448–3400 cm^−1^ corresponding to -OH, -NH-, and -NH_2_ groups. Additionally, two singlet signals were visible in the ^1^H-NMR spectrum at δ 5.22 and 6.02 ppm, attributed to two -NH_2_ groups.

Likewise, the pyrazole derivative **14** was prepared by reaction of α,β-unsaturated ketone **2** and carbohydrazide in boiling methanol and pyridine. ^1^H-NMR spectrum exhibited two singlet signals for the CH-_pyrazole_ and NH_2_ groups at δ 5.23 and 6.89 ppm. Finally, compound **2** also reacted with thiosemicarbazide in refluxing sodium ethoxide as a basic medium to furnish thioamide derivative **15 [28]**, as declared in Figure 4.

The spectroscopic data of all prepared molecules are included in the Appendix A as Appendix A.

### 2.2. In Vitro Inflammation Screening

#### 2.2.1. Protein Denaturation Inhibition

A repeated measures two-way ANOVA analysis comparing the synthesized molecules with diclofenac revealed a statistically significant interaction between the molecules across varying concentrations (*p* < 0.0001). To further explore the impact of each molecule, a multiple *t*-test was conducted, comparing each concentration of the synthesized compounds directly with the corresponding concentration of diclofenac. The majority of the synthesized compounds demonstrated significant anti-inflammatory activity, as summarized in Table 1. The findings indicated that all synthesized compounds exhibited substantial anti-inflammatory effects. Among them, compounds **11**, **12**, and **14** showed the most pronounced activity, followed by compounds **2**, **4**, **6**, **7**, **8**, **10**, **13**, and **15** which had percent range from 99% to 90%, and then compounds **3**, **5**, and **9** which had percent range less than 90%, across all tested concentrations when compared to the reference drug, diclofenac.

#### 2.2.2. Cell Membrane Protection Assay

For assessment of the statistical interaction between the prepared molecules at various concentrations, a repeated measures two-way ANOVA was performed. The analysis revealed a highly significant interaction between the molecules (*p* < 0.0001). Furthermore, the interaction of each molecule with the reference drug, diclofenac, was evaluated using multiple *t*-tests. The results, as detailed in Table 2, indicated that most of the compounds exhibited significant differences. Overall, all synthesized compounds demonstrated a notable percentage of membrane protection compared to diclofenac. Specifically, compounds **4**, **11**, and **15** exhibited the highest protection percentages across concentrations of 50, 100, 250, and 500 μg/mL, while compounds **2, 3, 8, 9**, and **12** showed the highest protection percentages at concentrations of 100 and 250 μg/mL.

#### 2.2.3. Quantitative Gene Expression Assessment

The results of the real-time PCR assay revealed a highly statistically significant interaction between the prepared molecules at various concentrations, as determined by RM two-way ANOVA (*p* < 0.0001), with beta-actin serving as the internal control. Most compounds exhibited significant differences. Notably, compound **12** demonstrated the most pronounced reduction in COX-2 expression levels, followed by compounds **6, 11**, and **13**, particularly at a concentration of 50 μg/mL (Figure 2).

### 2.3. Computational Studies

To better understand the binding mode of molecules against the target enzyme COX-2, in silico molecular docking studies were performed using PyRx-0.8 tool. Out of fourteen biologically tested molecules, four derivatives (**6**, **11**, **12**, and **13**) were selected for the docking studies. The obtained results are portrayed in Table 3. However, analysis of the docking results exhibited that the molecules displayed superior binding energies of −10.9 to −9.8 kcal/mol to the target enzyme, which are more favorable than the standard drug (−6.5 kcal/mol). Chemical interaction behaviors were performed between the molecules and COX-2 active site residues through non-covalent bonds such as hydrogen bonding and arene stacking. The amino acid residues such as ASP125, PRO127, TYR130, ASP133, TYR134, GLY135, VAL155, PRO156, and ASP157 were involved in the intermolecular interactions with the docked molecules [29]. The finding showed that diclofenac exhibited two H-bonds (HBs) and one arene–sigma interaction with ARG120 and TYR115 at 2.16, 1.96, and 2.76 Å, respectively. Molecule **6** docked with the active pocket of the target via two HBs and one arene–arene interaction at distances 2.46, 2.42 and 5.91 Å, respectively. The hydroxyl and amino groups are included in the hydrogen bonding interactions, while the phenyl moiety is included in the arene–arene interaction with TYR136. While molecule **11** docked with the target via one HB with PRO154 at 1.91 Å. The docking process was carried out through the hydroxyl group of the molecule with the residue PRO154. Further, compound **12** with the highest binding score (−10.9 kcal/mol) exhibited the ability to form various advantageous interactions such as one HB and one arene–arene stacking with the amino acid residues CYS47 and TYR136 at 2.22 and 5.72 Å, respectively. The hydroxyl group and phenyl moiety are included in the interactions. Finally, molecule **13** showed two HBs interactions with PRO154 and ASP157. The -CONHNH_2_ and amino groups in compound **13** are involved in the intermolecular interactions with the active site pockets of the target enzyme. The 2D and 3D interactions of the docked complexes are represented in Figure 3. The Structure–Activity Relationship (SAR) analysis declares that the presence of specific functional groups plays an important role in enhancing the anti-inflammatory activity of the molecules. In the case of compounds **6**, **11**, **12,** and **13**, which demonstrated significant suppression of COX-2 expression at 50 μg/mL, their functional groups contribute to their potency as follows:The electron-donating groups such as hydroxyl (-OH) [30] and methoxy (-OCH_3_) enhance the hydrogen bonding interactions with the binding site pockets of COX-2, leading to stronger enzyme inhibition. In addition, they also improve the solubility and bioavailability, increasing cellular uptake and efficacy.The presence of the amino group enhances the binding affinity to COX-2 by forming additional hydrogen bonding with the active site residues. Further, it also influences lipophilicity.The -CONHNH_2_ moiety provides further hydrogen bonding sites, enhancing the overall inhibitory mechanism.The aromatic systems contribute to arene–arene stacking with hydrophobic residues in COX-2, increasing their affinity and inhibitory potential.

On the other hand, the physicochemical and pharmacokinetic properties of molecules were predicted by means of SwissADME and admetSAR 2.0 free website. The molecular weights of molecules are less than 500 g/mol, showing good absorption. Consequently, they obeyed Lipinski’s rule of five without any violation. Their rotatable bonds are within the allowed range (<8 bond), indicating their flexibility. To conclude, we can say that compounds are predicted to have acceptable bioavailability.

## 3. Materials and Methods

### 3.1. Chemistry

All melting points were calculated without correction using the Kofler Block apparatus. IR spectral data (ν, cm^−1^) were recorded on an FTIR 5300 spectrometer (JASCO, Oklahoma City, OK, USA). The NMR spectral data were calculated on a Varian Gemini spectrometer (Varian, Inc., Lake Forest, CA, USA); 300 and 400 MHz for ^1^H-NMR and 100 MHz for ^13^C-NMR (DMSO-d*_6_* as solvent and Tetramethylsilane (TMS) as an internal reference). The GCMS-QP 1000 EX spectrometer (Shimadzu, Kyoto, Japan) at 70 eV was utilized to detect the mass spectral data. Using n-hexane and EtOAc, thin layer chromatography (TLC) was utilized to investigate the purity of the obtained molecules (9:1, V/V, 7:3 V/V) eluent. The elemental analyses were performed at Microanalytical Research Centers, Faculties of Science and Pharmacy, Cairo University, Egypt.

**1-(5-hydroxy-1,3-diphenyl-1H-pyrazol-4-yl)-3-(4-methoxyphenyl)prop-2-en-1-one (2):** 30 mL of a 10% ethanolic sodium hydroxide solution was used to suspend compound **1** (0.5 g; 1.79 mmol) and *p*-methoxybenzaldehyde (0.24 g; 1.76 mmol). The mixture was then agitated at room temperature for 24 h before refluxing for 1 hr. 50 mL of cold water was added once the solution had stopped. The formed precipitate was filtered out, water-washed, and then recrystallized from DMF/EtOH to furnish orange crystals (**2**; 0.57 g; 75%); m.p. > 300 °C. IR (KBr) ν cm^−1^ = 3393 (OH), 3061 (CH-_aromatic_), 2955 (CH-_aliphatic_), 1711 (C=O); ^1^H-NMR (300 MHz, DMSO-d_6_) ™ = 3.68 (s, 3H, OCH_3_), 5.12–5.15 (d, 1H, =CH-_olefinic_, *J* = 9 Hz), 6.75–6.78 (d, 1H, O=C-CH=, *J* = 9 Hz), 7.10–7.38 (m, 14H, Ar-H), 11.00 (s, 1H, OH); ^13^C-NMR (100 MHz, DMSO-d_6_) ™ =55.89, 114.27, 114.27, 114.98, 119.91, 123.43, 123.43, 123.43, 128.36, 129.03, 129.38, 129.38, 129.72, 129.72, 130.01, 130.01, 130.01, 130.01, 132.28, 136.90, 139.72, 153.01, 163.30, 167.47, 191.79; MS: *m/z* (%) 396 (M^+^). Anal. calcd. For C_25_H_20_N_2_O_3_ (396): C, 75.74; H, 5.08; N, 7.07; Found: C, 75.80; H, 5.10; N, 7.1%.

**2-hydroxy-6-(5-hydroxy-1,3-diphenyl-1H-pyrazol-4-yl)-4-(4-methoxyphenyl)-4H-pyran-3-carbonitrile (3):** For around 12 h, a suspension of compound **2** (0.5 g; 1.26 mmol) and ethyl cyanoacetate (0.143 g; 1.26 mmol) was refluxed in 30 mL of sodium ethoxide solution. Half of the solution’s volume was concentrated. After which it was allowed to stop and put into 50 mL of cool water, acidified with 2 drops of HCl. The formed precipitate was collected by filtration, then recrystallized to furnish brown crystals (**3**; 0.5 g; 73%); m.p. 148–150 °C; IR (KBr) ν cm^−1^ = 3449 (OH), 3061 (CH-_aromatic_), 2958 (CH-_aliphatic_), 2218 (CN); ^1^H-NMR (400 MHz, DMSO-d_6_) ™ = 3.71 (s, 3H, OCH_3_), 5.22 (s, 1H, 4H-_Pyran_), 6.02 (s, 1H, 5H-_Pyran_), 6.86–7.85 (m, 14H, Ar-H), 11.80 (s, 1H, OH), 14.57 (s, 1H, OH); ^13^C-NMR (100 MHz, DMSO-d_6_) ™ =32.71, 54.95, 55.76, 100.00, 113.76, 114.97, 114.97, 115.46, 121.11, 121.11, 127.93, 128.51 128.60, 128.60, 128.88, 128.88, 129.38, 129.38, 129.41, 129.41, 133.50, 133.50, 134.15, 139.28, 154.96, 157.59, 157.59, 196.03; MS = *m/z* (%) 463 (M^+^). Anal. calcd. For C_28_H_21_N_3_O_4_ (463): C, 72.56; H, 4.57; N, 9.07; Found: C, 73.62; H, 4.62; N, 9.13%.

**6-(5-hydroxy-1,3-diphenyl-1H-pyrazol-4-yl)-4-(4-methoxyphenyl)-2-oxo-1,2-dihydropyridine-3-carbonitrile (4):** Absolute ethanol (30 mL) was used to reflux a solution of compound **2** (0.5 g; 1.26 mmol), ethyl cyanoacetate (0.143 g; 1.26 mmol), and ammonium acetate (2 g) for 12 h. The solution was concentrated to half its original volume, stopped, and then poured into 50 mL of crushed ice and water. The resulting solid residue was filtered and then crystallized from ethanol to yield yellow crystals (**4**; 1.9 g; 72%); m.p. 180–182 °C. IR (KBr) ν cm^−1^ = 3448–3400 (OH/NH), 3062 (CH-_aromatic_), 2932 (CH-_aliphatic_), 2209 (CN), 1707 (C=O); ^1^H-NMR (300 MHz, DMSO-d_6_) ™ = 3.70 (s, 3H, OCH_3_), 6.60–8.07 (m, 15H, Ar-H+ H-_pyridone_), 9.90 (hump, 1H, NH), 11.00 (hump, 1H, OH);^13^C-NMR (100 MHz, DMSO–d_6_) ™ = 55.32, 81.18, 100.00, 113.81, 113.81, 118.79, 119.12, 124.81, 124.81, 126.97, 128.03, 128.03, 128.03, 128.67, 129.43, 129.43, 129.74, 129.74, 130.90, 130.90, 131.10,137.77, 150.09, 157.90, 158.35, 158.75, 159.02, 174.86; MS = *m/z* (%) 460 (M^+^). Anal. calcd. For C_28_H_20_N_4_O_3_ (460): C, 73.03; H, 4.38; N, 12.17; Found: C, 73.1; H, 4.40; N, 12.37%.

**2-(3-(5-hydroxy-1,3-diphenyl-1H-pyrazol-4-yl)-1-(4-methoxyphenyl)-3-oxopropyl)malononitrile (5):** For about 30 min, the combination of compound **2** (0.5 g; 1.26 mmol**)** and malononitrile (0.083 g; 1.25 mmol) was refluxed in 50 mL of glacial acetic acid. After that, the resulting solution was allowed to cool and was concentrated to half its volume. The separated solid was removed using filtering, rinsed with ethanol, and then crystallized from ethanol to yield reddish-brown crystals (**5**; 0.37 g; 64%); m.p. 130–132 °C; IR (KBr) ν cm^−1^ = 3451 (OH), 3059 (CH-_aromatic_), 2931–2836 (CH-_aliphatic_), 2195 (CN), 1712 (C=O); ^1^H-NMR (400 MHz, DMSO-d_6_) ™ = 3.30–3.32 (d, 2H, CH_2_, *J* = 8.9 Hz), 3.71–3.72 (q, 1H, CH-Ph, *J* = 5.2 Hz), 3.82 (s, 3H, OCH_3_), 5.24–5.30 (d, 1H, CH-(CN)_2_, *J* = 24.4 Hz), 6.87–7.84 (m, 14H, Ar-H), 14.29 (br. s, 1H, OH); ^13^C-NMR (100 MHz, DMSO–d_6_) ™ = 24.80, 32.67, 39.92, 55.38, 113.78, 113.78, 113.84, 113.84, 120.94, 121.64, 121.64, 126.20, 127.82, 127.82, 127.93, 127.93, 128.51, 129.08, 129.08, 129.14, 129.14, 131.33, 133.35, 137.52, 148.99, 157.70, 157.70, 177.02; MS = *m/z* (%) 462 (M^+^). Anal. calcd. For C_28_H_22_N_4_O_3_ (462): C, 72.71; H, 4.79; N, 12.11; Found: C, 72.77; H, 4.85; N, 12.13%.

**2-amino-6-(5-hydroxy-1,3-diphenyl-1H-pyrazol-4-yl)-4-(4-methoxyphenyl)-4H-pyran-3-carbonitrile (6): Method A**: After dissolving compound **10** (0.5 g) in 20 mL of glacial acetic acid and adding 1 g of sodium acetate, the resultant solution was refluxed for 6 h. The solution was allowed to stop before being poured into 50 mL of ice-cold water, acidified with 2 drops of HCl. The separated solid was filtered off after being cleaned with water and then recrystallized from DMF/EtOH to give yellow crystals (**6**; 0.41 g; 75%). **Method B:** Malononitrile (0.083 g; 1.25 mmol) and compound **2** (0.5 g; 1.26 mmol) were combined and refluxed in sodium ethoxide solution for 12 h. After allowing the mixture to cool, it was added to a cold diluted HCl. After filtration, the formed precipitate was recrystallized from DMF/ethanol to provide (**6**; 0.32 g; 55%) as yellow crystals; m.p. 280–282 °C; IR (KBr) ν cm^−1^ = 3448–3300 (OH/NH_2_), 3063 (CH-_aromatic_), 2924 (CH-_aliphatic_), 2218 (CN); ^1^H-NMR (400 MHz, DMSO-d_6_) ™ =3.84 (s, 3H, OCH_3_), 4.60 (s, 1H, 4H-_pyran_), 6.04 (s, 2H, NH_2_), 7.27–7.83 (m, 15H, Ar-H+ H-_pyran_), 14.29 (hump, 1H, OH); ^13^C-NMR (100 MHz, DMSO–d_6_) ™ = 21.01, 55.31, 58.15, 85.87, 97.50, 100.08, 114.49, 114.49, 119.70, 122.20, 122.20, 125.60, 126.26, 128.44, 128.44, 129.05, 129.05, 129.08, 129.08, 129.38, 129.38, 133.54, 137.08, 139.08, 150.03, 154.61, 157.03, 162.02; MS = *m/z* (%) 463 (M^+^+1). Anal. calcd. For C_28_H_22_N_4_O_3_ (462):C, 72.71; H, 4.79; N, 12.11; Found: C, 72.77; H, 4.85; N, 12.15%.

**2-amino-6-(5-hydroxy-1,3-diphenyl-1H-pyrazol-4-yl)-4-(4-methoxyphenyl)nicotine nitrile (7):** A solution of glacial acetic acid (15 mL) containing ammonium acetate (2 g; 0.026 mmol) and a combination of compound **2 (**0.5 g; 1.26 mmol**)** and malononitrile (0.083 g; 1.26 mmol) was refluxed for approximately 12 h. Half of its initial volume was extracted from the solution. Pour the cooled mixture into 50 milliliters of cold water. The formed precipitate was collected by filtration, then recrystallized from ethanol to afford brown crystals (**7**; 1.56 g; 60%); m.p. 144–146 °C; IR (KBr) ν cm^−1^ = 3450–3373 (NH_2_/OH), 3062 (CH-_aromatic_), 2930 (CH-_aliphatic_), 2202 (CN); ^1^H-NMR (400 MHz, DMSO-d_6_) ™ = 3.86 (s, 3H, OCH_3_), 6.02 (s, 2H, NH_2_), 6.64–8.09 (m, 15H, Ar-H+ H-_pyridone_), 11.35 (s, 1H, OH); MS = *m/z* (%) 460 (M^+^+1). Anal. calcd. For C_28_H_21_N_5_O_2_ (459): C, 73.19; H, 4.61; N, 15.24; Found: C, 73.26; H, 4.66; N, 15.29%.

**4-(4-(4-methoxyphenyl)-1,4,5,6,7,8-hexahydroquinolin-2-yl)-1,3-diphenyl-1H-pyrazol-5-ol (8):** Compound **2** (0.5 g, 1.26 mmol) and cyclohexanone (0.13 g, 1.26 mmol) were combined and refluxed for roughly 24 h in 20 mL of glacial acetic acid that also included 1 g of ammonium acetate. Half of the solution’s volume was concentrated. After cooling and being put into 50 mL of cold water, the formed precipitate was collected by filtration, then recrystallized from ethanol to produce yellow crystals (**8**; 1.25 g; 77%); m.p. 170–172 °C; IR (KBr) ν cm^−1^ = 3401–3350 (OH/NH), 3059 (CH-_aromatic_), 2929–2857 (CH-_aliphatic_); ^1^H-NMR (400 MHz, DMSO-d_6_) ™ = 1.96–2.25 (m, 4H, 2CH_2_), 2.26-2.97 (m, 4H, 2CH_2_), 3.80 (s, 3H, OCH_3_), 5.34 (s, 1H,CH-_pyridine_), 6.61-8.08 (m, 15 H, Ar-H), 10.30 (s, 1H, NH), 14.80 (s, 1H, OH); ^13^C-NMR (100 MHz, DMSO-d_6_) ™ = 22.53, 24.80, 25.18, 31.86, 42.10, 55.31, 97.70, 111.10, 113.79, 113.79, 119.10, 122.94, 122.94, 126.95, 127.50, 127.50, 129.35, 129.35, 129.41, 129.41, 129.70, 130.88, 130.88, 133.50, 134.70, 136.60, 137.03, 139.01, 157.11, 158.34, 158.75; MS = *m/z* (%) 477 (M^+^+2). Anal. calcd. ForC_31_H_29_N_3_O_2_ (475): C, 78.29; H, 6.15; N, 8.84; Found: C, 78.36; H, 6.20; N, 8.90%.

**3-acetyl-6-(5-hydroxy-1,3-diphenyl-1H-pyrazol-4-yl)-4-(4-methoxyphenyl)pyridin-2(1H)-one (9):** Compound **2** (0.5 g; 1.26 mmol) and ethyl acetoacetate (0.16 g; 1.23 mmol) were combined and refluxed for hrs in glacial acetic acid (20 mL) in the presence of ammonium acetate (1 g). After the solution was reduced to half of its original volume and allowed to cool, it was added to 50 milliliters of ice and water. The formed precipitate was collected by filtration, then recrystallized from ethanol to yield yellow crystals (**9**; 1.16 g; 70%); m.p. 122–124 °C; IR (KBr) ν cm^−1^ = 3423–3400 (OH/NH), 3061 (CH-_aromatic_), 2927 (CH-_aliphatic_), 1720,1601 (2C=O); ^1^H-NMR (400 MHz,DMSO-d_6_) ™ = 2.04 (s, 3H, COCH_3_), 3.85 (s, 3H, OCH_3_), 7.08–8.05 (m, 15H, Ar-H+ H-_pyridone_), 10.80 (s, 1H, NH), 11.00 (s, 1H, OH); MS = *m/z* (%) 477 (M^+^). Anal. calcd. ForC_29_H_23_N_3_O_4_ (477):C, 72.94; H, 4.85; N, 8.80; Found: C, 72.99; H, 4.88; N, 8.85%.

**1-(2-hydroxy-6-(5-hydroxy-1,3-diphenyl-1H-pyrazol-4-yl)-4-(4-methoxyphenyl)-4H-pyran-3-yl)ethanone (10):** Sodium ethoxide solution (20 mL) was added to a solution of compound **2** (0.5 g; 1.26 mmol) and ethyl acetoacetate (0.164 g; 1.26 mmol) and heated for approximately 24 h. After letting the solution cool, it was added to 50 milliliters of ice and water and acidified with two drops of hydrochloric acid. After being removed from the ethanol, the material was filtered out, and recrystallized from ethanol to furnish yellow powder (**10**; 0.53 g; 80%); m.p. 150–152 °C; IR (KBr) ν cm^−1^ = 3402 (OH), 3059 (CH-_aromatic_), 2931 (CH-_aliphatic_), 1720 (C=O); ^1^H-NMR (400 MHz,DMSO-d_6_) ™ = 2.00 (s, 3H, COCH_3_), 3.70 (s, 3H, OCH_3_), 5.23 (s, 1H, 4H-_pyran_), 6.01 (s, 1H, 5H-_pyran_), 6.87–7.83 (m, 14H, Ar-H), 10.38 (s, 1H, OH), 11.80 (s, 1H, OH);^13^C-NMR (100 MHz, DMSO-d_6_) ™ = 22.65, 32.64, 54.99, 85.20, 113.62, 113.76, 113.83, 113.83, 121.28, 121.28, 126.12, 127.83, 127.83, 128.45, 128.95, 128.95, 129.17, 129.17, 133.18, 133.18, 133.31, 133.39, 138.82, 138.82, 149.58, 157.67, 157.67, 171.17, 196.73; MS = *m/z* (%) 480 (M^+^). Anal. calcd. For C_29_H_24_N_2_O_5_ (480): C, 72.49; H, 5.03; N, 5.83; Found: C, 72.52; H, 5.05; N, 5.86%.

**4-(5-(4-methoxyphenyl)-4,5-dihydro-1H-pyrazol-3-yl)-1,3-diphenyl-1H-pyrazol-5-ol (11):** A suspension of compound **2** (0.5 g; 1.26 mmol) and hydrazine hydrate (0.06 g; 1.26 mmol) was heated under reflux for a 6 h period in absolute ethanol (25 mL). Half of the mixture’s original volume was concentrated. After adding two drops of HCl to 50 milliliters of cold water, the resulting solution was added. The formed precipitate was collected by filtration, then recrystallized from ethanol to provide reddish-brown powder (**11**; 0.42 g; 75%); m.p. 133–135 °C; IR (KBr) ν cm^−1^ = 3300–3188 (OH/NH), 3058 (CH-_aromatic_), 2955 (CH-_aliphatic_); ^1^H-NMR (400 MHz, DMSO-d_6_) ™ = 3.67 (s, 3H, OCH_3_), 3.73 (d, 2H, CH_2_-_pyrazole_, *J* = 20 Hz), 3.80 (t, 1H, CH-_pyrazole_, *J* = 16 Hz), 6.61–8.18 (m, 14H, Ar-H),11.48 (s, 1H, OH), 11.86 (s, 1H, NH); ^13^C-NMR (100 MHz, DMSO-d_6_) ™ = 39.92, 54.50, 55.84, 112.34, 113.32, 114.36, 114.36, 124.95, 125.99, 126.48, 127.12, 127.28, 127.28, 128.96, 129.03, 129.03, 129.96, 129.96, 131.59, 138.10, 138.37, 142.90, 146.33, 156.92, 163.63; MS = *m/z* (%) 410 (M^+^). Anal. calcd. For C_25_H_22_N_4_O_2_ (410): C, 73.15; H, 5.40; N, 13.65; Found: C, 73.22; H, 5.45; N, 13.68%.

**1-(3-(5-hydroxy-1,3-diphenyl-1H-pyrazol-4-yl)-5-(4-methoxyphenyl)-4,5-dihydro-1H-pyrazol-1-yl)ethanone (12): Method A:** For 24 h, a solution of compound **2** (0.5 g; 1.26 mmol) and hydrazine hydrate (0.06 g; 1.26 mmol) was boiled in glacial acetic acid (20 mL). Half of the solution’s original volume was concentrated. After that, we let it cool. After being filtered out, the separated product was cleaned with ethanol and crystallized in dioxane to give yellow crystals (**12**, 0.42 g; 76%). **Method B:** For 12 h, a sample of compound **11** (0.5 g) in (20 mL) of glacial acetic acid was fused, letting the reaction mixture cool until it has reduced in volume by half. Following filtering, a water wash, and crystallization from dioxane, the separated solid was obtained (**12**; 0.34 g; 68%) as white crystals; m.p. 190–192 °C; IR (KBr) ν cm^−1^ = 3360 (OH), 3042 (CH-_aromatic_), 2957 (CH-_aliphatic_), 1704 (C=O); ^1^H-NMR (400 MHz,DMSO-d_6_) ™ = 1.87 (s, 3H, COCH_3_), 3.36 (d, 2H, CH_2_-_pyrazole_, *J* = 9.6 Hz), 3.60 (t, 1H, CH-_pyrazole_, *J* = 25.6 Hz), 3.70 (s, 3H, OCH_3_), 6.82–7.93 (m, 14H, Ar-H), 11.00 (s, 1H, OH); MS = *m/z* (%) 452 (M^+^). Anal. calcd. ForC_27_H_24_N_4_O_3_ (452): C, 71.67; H, 5.35; N, 12.38; Found: C, 71.74; H, 5.40; N, 12.40%.

**2-amino-6-(5-hydroxy-1,3-diphenyl-1H-pyrazol-4-yl)-4-(4-methoxyphenyl) nicotine hydrazide (13):** Compound **2** (0.5 g; 1.26 mmol) and 2-cyanoacetohydrazide (0.123 g; 1.24 mmol) were suspended in a glacial acetic acid solution (15 mL) containing ammonium acetate (2 g; 25.9 mmol) and refluxed for approximately 12 h. Half of the solution’s volume was concentrated. After letting the reaction solution cool and adding it to 50 milliliters of cold water, the formed precipitate was collected by filtration, then recrystallized from ethanol to furnish pale yellow powder (**13**; 2.2 g; 84%); m.p. 140–142 °C; IR (KBr) ν cm^−1^ = 3448–3400 (OH/NH_2_/NH), 3060 (CH-_aromatic_), 2925 (CH-_aliphatic_), 1660 (C=O); ^1^H-NMR (400 MHz,DMSO-d_6_) ™ = 3.84 (s, 3H, OCH_3_), 5.22 (s, 2H, NH_2_), 6.02 (s, 2H, NH_2_), 7.25–7.92 (m, 15H, Ar-H+ H-_pyridine_), 10.50 (s, 1H, NH), 11.00 (s, 1H, OH); MS = *m/z* (%) 492 (M^+^). Anal. calcd. For C_28_H_24_N_6_O_3_ (492):C, 68.28; H, 4.91; N, 17.06; Found: C, 68.31; H, 4.93; N, 17.09%.

**(5-amino-1,3-diphenyl-1H-thieno [3,2-c]pyrazol-6-yl)(5′-hydroxy-5-(4-methoxyphenyl)-1′,3′-diphenyl-1H,1′H-3,4′-bipyrazol-1-yl)methanone (14):** Compound **2** (0.5 g; 1.26 mmol) and 5-amino-1,3-diphenyl-1H-thieno [3,2-c]pyrazole-6-carbohydrazide (0.44 g; 1.26 mmol), a catalytic amount of pyridine (2 mL) in 30 mL of methanol, was heated under reflux for approximately 10 h. After letting the solution cool, it was poured into 50 milliliters of diluted HCl. After the separated substance was filtered, water-cleaned, and methanol was crystallized to give yellow crystals (**14**; 0.62 g; 66%); m.p. 310–312 °C; IR (KBr) ν cm^−1^ = 3449–3400 (OH/NH_2_), 3060 (CH-_aromatic_), 2931 (CH-_aliphatic_), 1601(C=O); ^1^H-NMR (400 MHz, DMSO-d_6_) ™ = 3.87 (s, 3H, OCH_3_), 5.23 (s, 1H, CH-_pyrazole_), 6.89 (s, 2H, NH_2_), 7.10-7.92 (m, 24H, Ar-H), 14.29 (hump, 1H, OH); MS = *m/z* (%) 727 (M^+^+2). Anal. calcd. ForC_43_H_31_N_7_O_3_S (725): C, 71.16; H, 4.30; N, 13.51; Found: C, 71.18; H, 4.32; N, 13.53%.

**3-(5-hydroxy-1,3-diphenyl-1H-pyrazol-4-yl)-5-(4-methoxyphenyl)-4,5-dihydro-1H-pyrazole-1-carbothioamide (15):** Thiosemicarbazide (0.115 g; 1.26 mmol) and compound **2** (0.5 g; 1.26 mmol) were added to a sodium ethoxide solution (30 mL) and heated for nearly 8 h while in reflux. After letting the contents cool, it was poured over broken ice and acidified with HCl. Filtered, rinsed with water, and recrystallized from EtOH to furnish brown crystals (**15**; 0.42 g; 69%); m.p. 120–122 °C; IR (KBr) ν cm^−1^ = 3449–3400 (OH/NH_2_), 3059 (CH-_aromatic_), 2955–2927 (CH-_aliphatic_); ^1^H-NMR (400 MHz,DMSO-d_6_) ™ = 3.71 (d, 2H, CH_2_-_pyrazole_, *J* = 13.2 Hz), 3.87 (s, 3H, OCH_3_), 3.90 (t, 1H, *sp*^3^ CH-_pyrazole,_
*J* = 21.6 Hz), 6.87 (s, 2H, NH_2_), 6.89–7.92 (m, 14H, Ar-H), 14.29 (hump, 1H, OH); ^13^C-NMR (100 MHz, DMSO-d_6_) ™ = 40.04, 55.25, 85.23, 113.84, 114.13, 114.13, 121.30, 121.30, 126.17, 126.17, 126.73, 127.89, 127.89, 128.10, 128.87, 128.87, 128.95, 128.95, 133.35, 133.35, 138.31, 142.20, 149.57, 153.92, 157.67, 177.60; MS = *m/z* (%) 469 (M^+^). Anal. calcd. ForC_26_H_23_N_5_O_2_S (469):C, 66.50; H, 4.94; N, 14.91; Found: C, 66.57; H, 4.98; N, 14.94%.

### 3.2. In Vitro Inflammation Screening

#### 3.2.1. Protein Denaturation Inhibition

The anti-inflammatory efficacy of the synthesized compounds was assessed by evaluating their ability to prevent bovine serum albumin (BSA) denaturation [31]. The experimental setup involved preparing a reaction mixture consisting of 50 μL of the test compound at different concentrations (500, 250, 100, and 50 μg/mL), combined with 450 μL of a 1% aqueous solution of BSA. The pH was carefully adjusted to 6.4 using 1 N HCl. Distilled water was used as a negative control, whereas diclofenac potassium served as the reference standard for comparison. The reaction mixtures were initially incubated at 37 °C for 20 min, followed by heat treatment at 60 °C for 15 min to induce protein denaturation. After cooling, the turbidity was measured at 600 nm to determine the extent of protein stabilization. All tests were conducted in triplicate, and the average values were used for analysis. The inhibition of protein denaturation was calculated using the following formula:Percent inhibition = (Abs control − Abs sample) × 100/Abs control. 

#### 3.2.2. Cell Membrane Protection Assay

The protective effect of the synthesized compounds on red blood cell (RBC) membranes was evaluated to assess their anti-inflammatory potential [31]. For this assay, human erythrocytes were isolated from a healthy volunteer who had refrained from taking NSAIDs for at least two weeks before the experiment. The reaction mixture contained an erythrocyte suspension and a hypotonic solution (50 mM NaCl in 10 mM sodium phosphate buffer, pH 7.4). Distilled water was used as a negative control, while diclofenac potassium was used as the standard reference drug at concentrations of 50, 100, 250, and 500 μg/mL. The samples were incubated at 37 °C for 30 min, followed by centrifugation at 3000 rpm for 20 min to separate intact erythrocytes from hemolyzed cells. The amount of released hemoglobin in the supernatant was then measured at 560 nm to determine the extent of membrane stabilization. Each experiment was conducted in triplicate, and the results were analyzed as mean values. The percentage of RBC membrane stabilization (protection) was determined using the following formula:Percent protection = 100 − (Abs sample/Abs control) 

#### 3.2.3. Quantitative Gene Expression Assessment

Peripheral blood mononuclear cells (PBMCs) were isolated from healthy donors who had refrained from NSAIDs for at least four days before blood collection. Cells were seeded in a 24-well plate at a density of 1 × 10^5^ cells per 500 μL of culture medium [RPMI 1640 (Biowest), supplemented with 10% heat-inactivated fetal bovine serum (FBS), 1% penicillin/streptomycin, 1% fungi zone solution (LONZA), and 1% HEPES buffer (1 M)]. The cells were incubated for 1 h at 37 °C in a 5% CO_2_ atmosphere. Following this, each test compound at a concentration of 50 μg/mL was added to the respective wells. Diclofenac potassium was used as the reference standard at the same concentration [31]. The plate was incubated for 24 h under the same conditions. The cultured PBMCs were then harvested, and RNA extraction was performed using an RNA extraction kit (Biovision, Inc., Milpitas, CA, USA). qPCR was conducted using the Novo™ cDNA Kit (Biovision, Inc.) and SYBR Green master mix (Thermo Fisher Scientific, Waltham, MA, USA). The cycling parameters were set to 95 °C for 10 min, followed by 50 cycles of 95 °C for 20 s and 55 °C for 30 s [32]. The relative expression of the COX-2 gene was evaluated using cDNA as a template, with β-actin as the endogenous control. SYBR Green master mix (Transgen: AQ601-01) was used, and the ΔΔCT method was applied and StepOne™ Real-Time PCR System (48-well). The following primer sequences were designed for each gene as shown in Table 4. The ratio of expression of each gene was represented as the mean of three experiments. The gene expression levels were identified by the relative comparative quantitation method.

Peripheral blood mononuclear cells (PBMCs) were obtained from healthy donors who had abstained from NSAID use for at least four days before blood collection. The isolated cells were cultured in a 24-well plate at a density of 1 × 10^5^ cells per 500 μL of RPMI 1640 medium. The culture medium was supplemented with 10% heat-inactivated fetal bovine serum (FBS), 1% penicillin/streptomycin, 1% fungizone solution, and 1% HEPES buffer. The cells were incubated at 37 °C in a 5% CO_2_ atmosphere for one hour to allow proper adherence and stabilization. Following incubation, 50 μg/mL of each test compound was introduced into designated wells. Diclofenac potassium was included as a reference standard at the same concentration. The plate was further incubated for 24 h under the same conditions. After incubation, the treated PBMCs were harvested, and total RNA was extracted using an RNA extraction kit (Biovision, Inc). Quantitative real-time PCR (qPCR) was performed using the Novo™ cDNA Kit (Biovision, Inc.) and SYBR Green Master Mix (Thermo Fisher Scientific). The cycling conditions were set as follows: Initial denaturation at 95 °C for 10 min, 50 cycles of 95 °C for 20 s, and annealing at 55 °C for 30 s. The COX-2 gene expression was measured using the ΔΔCT method, with β-actin as an internal control. The qPCR analysis was conducted using the StepOne™ Real-Time PCR System (48-well). The primer sequences used for the COX-2 and β-actin genes are listed in Table 5. The final gene expression levels were calculated as the mean of three independent experiments using the relative quantification approach.

#### 3.2.4. Statistical Analysis

The data are presented as mean ± standard deviation (SD). Statistical analysis was performed using GraphPad Prism 8 (San Diego, CA, USA), applying multiple *T*-tests and two-way ANOVA to assess significance. A *p*-value of <0.05 was considered statistically significant, while *p* < 0.001 was classified as highly significant, and *p* < 0.0001 was regarded as extremely significant.

### 3.3. Docking Studies

The crystal structure of the target enzyme COX-2 (PDBID: 5kir) [12] was obtained from the RCSB Protein Data Bank web server (www.rcsb.org/pdb/). The docking protocol was reported earlier by our group members [25]. The 2D structures of the synthesized compounds are generated by ChemDraw Professional 16.0 and saved as SDF files using the Open Babel 2.4.1 tool [33]. The target crystal structure and the compounds were further optimized, and energy minimized to identify the stable conformers, using CHARMM Force Field [34] in Discovery Studio and universal force field (UFF) [35] in Open Babel, respectively. The grid box of dimension 25 Å *25 Å *25 Å covers the active site of the target to ensure the docking with the compounds. Then, the docking process of the compounds and reference drug with the target was performed using PyRx 0.8—virtual screening tool [36]. Nine conformers for each docked compound were obtained, and the lowest energy confirmation was selected for further study. Also, the molecular visualization of the interactions was performed using Discovery Studio Visualizer 3.5. Using SwissADME (http://www.swissadme.ch) [37] and admetSAR 2.0 [38] websites, several physicochemical and pharmacokinetic properties of molecules were calculated.

## 4. Conclusions

In continuation of our ongoing work towards the design of new and promising anti-inflammatory agents, herein, we describe the synthesis, anti-inflammatory evaluation, and docking studies of some new heterocyclic compounds incorporating pyridine, pyran, and pyrazole moieties **3**–**15**. The newly prepared heterocyclic compounds were characterized by means of spectral analyses (IR, ^1^H-NMR, ^13^C-NMR, MS), along with elemental analyses. Further, the anti-inflammatory potential of these compounds was investigated in vitro using protein denaturation inhibition and cell membrane protection assays. Among them, compound **12**, followed by **13**, **11**, and **6**, exhibited potent anti-inflammatory properties. In addition, the docking studies were performed and revealed that most of the synthesized compounds exhibited good binding energy towards the target ranging from −10.9 to −9.8 kcal/mol, which are more potent than the reference drug (−6.5 kcal/mol). Structure Activity Relationship (SAR) suggested that the presence of various electron-donating groups such as –OH, –OCH_3_, and –NH_2_ markedly influenced the selectivity for COX-2. The obtained results from in silico studies were in consent with the biological work. The outcomes of both studies open new doors for designing new heterocyclic compounds containing pyridine, pyran, and/or pyrazole moieties as potential anti-inflammatory inhibitors.

## Data Availability

The original contributions presented in this study are included in the article/Appendix A. Further inquiries can be directed to the corresponding author.

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
