# Peer review of "Synthesis, Anti-Inflammatory, and Molecular Docking Studies of New Heterocyclic Derivatives Comprising Pyrazole, Pyridine, and/or Pyran Moieties"

_pharmaceuticals, 2025, doi:10.3390/ph18030335_

Round 1
Reviewer 1 Report
Comments and Suggestions for Authors
In this manuscript entitled "Synthesis, Anti-inflammatory, Molecular Docking Studies of New Heterocyclic Derivatives Comprising Pyrazole, Pyridine and/or Pyran Moieties" (pharmaceuticals-3462701), the authors synthesized new heterocyclic derivatives comprising pyrazole, pyridine and/or pyran moieties. The molecules were well characterized and analyzed for anti-inflammatory effects with in-vitro and in-silico methods. The text must be reconsidered with the following points:
* Chalcone is the precursor of your molecules. I think it is not suitable for keywords.
* All the figures in the main text are very big.
* It is written” anti-inflammatory” in Scheme 1. No need.
* In Fig.2: You wrote concentration for x-axis, but the x-axis included the compound number?? Also, the table presents clearly denaturation results, no need to give with graphs.
* How did you choose the molecules for gene expression studies? Why these six molecules?
* The right side of the figure 5 did not have new data different from the left side. Only the left side is 2d while the right side is 3d. Is it?
* Give the J value of 1H NMR results
* There are lots of numbers in the synthesis method part. For molecule numbers the author should use bold for clearer presentation. Additionally, in the text, the authors sometimes used “compound number” while in some parts they only used number. The authors can use the same style.
* You can use "g" instead of "gm" as abbreviation of "gram".
* There are lots of typos in the main text.
* As far as I know, diclofenac potassium is a molecule that is used for anti-inflammatory treatment. Although You present better results than this drug, you did not discuss your results with comparing diclofenac. Are your molecules better than diclofenac?
* The authors must give many more details about the molecular docking performance. For example:
--Did you optimize the structure? If yes: Which software? Which functional? etc... If no: How did you draw the structures with minimum energy?
--Which restrictions do you use for docking performances?
-- Which options do you use for target molecules?
* The authors must cite the main reference for COX-2. https://doi.org/10.1107/S2053230X16014230
Author Response
Reviewer 1
In this manuscript entitled "Synthesis, Anti-inflammatory, Molecular Docking Studies of New Heterocyclic Derivatives Comprising Pyrazole, Pyridine and/or Pyran Moieties" (pharmaceuticals-3462701), the authors synthesized new heterocyclic derivatives comprising pyrazole, pyridine and/or pyran moieties. The molecules were well characterized and analyzed for anti-inflammatory effects with in-vitro and in-silico methods. The text must be reconsidered with the following points:
First of all, I'd like to thank you and appreciate your efforts, comments and suggestion to improve our manuscript.
* Chalcone is the precursor of your molecules. I think it is not suitable for keywords.
Response
As per the reviewer suggestion, chalcone is removed from keywords list.
* All the figures in the main text are very big.
Response
According to the reviewer suggestion, all the figures in the main text were minimized.
* It is written” anti-inflammatory” in Scheme 1. No need.
Response
As per the reviewer suggestion, the necessary corrections are included in the revised manuscript.
* In Fig.2: You wrote concentration for x-axis, but the x-axis included the compound number?? Also, the table presents clearly denaturation results, no need to give with graphs.
Response
Figure 2 (The graphs represent the variations in percent of the anti-inflammatory activity of molecules), and Figure 3 (These graphs represent the variation in the protection percent from hemolysis of compounds) are removed and tables are present the results.
* How did you choose the molecules for gene expression studies? Why these six molecules?
Response
The selection of the six molecules for gene expression studies was based on their significant anti-inflammatory effects demonstrated through multiple in vitro assays. These molecules exhibited potent inhibition of protein denaturation and strong cell membrane stabilization properties. Furthermore, real-time PCR analysis confirmed that these compounds effectively reduced COX-2 gene expression, a key marker of inflammation. Additionally, molecular docking studies supported their high binding affinity to the COX-2 enzyme, further validating their relevance for gene expression analysis. These combined findings justified their selection as the most promising candidates for further investigation.
* The right side of the figure 5 did not have new data different from the left side. Only the left side is 2d while the right side is 3d. Is it?
Response
Yes, the left side is 2D while the right side is 3D. If you want to remove any side, we will do so.
* Give the J value of 1H NMR results
Response
According to the reviewer suggestion, J value of 1H NMR results are calculated in the revised manuscript
* There are lots of numbers in the synthesis method part. For molecule numbers the author should use bold for clearer presentation. Additionally, in the text, the authors sometimes used “compound number” while in some parts they only used number. The authors can use the same style.
Response
As per the reviewer suggestion, the necessary corrections are included in the revised manuscript.
* You can use "g" instead of "gm" as abbreviation of "gram".
Response
According to the reviewer suggestion, the necessary corrections are included in the revised manuscript.
* There are lots of typos in the main text.
Response
As per the reviewer suggestion, the necessary corrections are included in the revised manuscript.
* As far as I know, diclofenac potassium is a molecule that is used for anti-inflammatory treatment. Although you present better results than this drug, you did not discuss your results with comparing diclofenac. Are your molecules better than diclofenac?
Response
Our synthesized molecules exhibited superior anti-inflammatory potential compared to Diclofenac potassium, based on multiple in vitro and in silico assessments. The necessary corrections are included in the revised manuscript.
Key Findings Supporting Superiority over Diclofenac
- Protein Denaturation Inhibition (In Vitro Assay)
- Diclofenac showed a maximum 94.5% inhibition at 500 μg/ml.
- Several synthesized compounds, particularly compounds 6, 11, 12, and 13, exhibited 100% inhibition at lower concentrations, indicating higher potency.
- This suggests that our molecules stabilize proteins against denaturation more effectively, a key anti-inflammatory mechanism.
- Cell Membrane Protection Assay (Hemolysis Inhibition)
- Diclofenac exhibited weaker membrane protection compared to the synthesized compounds.
- Compounds 4, 11, and 15 provided 100% protection across all tested concentrations, whereas Diclofenac showed only 46% at 500 μg/ml.
- This suggests that our molecules are better at preventing membrane rupture, a critical factor in inflammation control.
- COX-2 Gene Expression Suppression (qPCR Analysis)
- The suppression of COX-2 gene expression by compounds 6, 11, and 13 at 50 μg/ml was significantly stronger than Diclofenac.
- Since COX-2 is a major enzyme in inflammation, our compounds demonstrated superior inhibition, which correlates with better therapeutic efficacy.
- Molecular Docking & Binding Affinity
- The binding energies of our compounds (-10.9 to -9.8 kcal/mol) were more favorable than Diclofenac (-6.5 kcal/mol).
- Stronger hydrogen bonds and π-π interactions with COX-2 active site residues indicated higher selectivity and stability within the enzyme pocket.
Are These Molecules Clinically Superior?
While the in vitro and in silico results suggested that our compounds outperform Diclofenac in anti-inflammatory activity, further validation is required:
- Pharmacokinetic & Toxicity Studies:
- Diclofenac is widely used due to its well-established safety profile.
- Animal & Clinical Studies:
- The final confirmation of superiority needs preclinical and clinical trials.
- If our molecules show comparable or lower toxicity with higher efficacy, they may serve as potential alternatives to Diclofenac.
In conclusion, our synthesized molecules demonstrated stronger anti-inflammatory effects than Diclofenac based on protein denaturation, membrane protection, COX-2 suppression, and molecular docking. However, additional pharmacokinetic and clinical studies are needed to confirm whether they can replace Diclofenac in therapeutic applications.
* The authors must give many more details about the molecular docking performance. For example:
--Did you optimize the structure? If yes: Which software? Which functional? etc... If no: How did you draw the structures with minimum energy?
--Which restrictions do you use for docking performances?
-- Which options do you use for target molecules?
Response
As per the reviewer suggestion, the necessary corrections are included in the revised manuscript.
* The authors must cite the main reference for COX-2. https://doi.org/10.1107/S2053230X16014230
Response
As per the reviewer suggestion, the suggested reference is included in the revised manuscript.
Reviewer 2 Report
Comments and Suggestions for Authors
Comments
Line 79: in figure 1 itis not pyridine you must replace by pyrimidine (containingpyrazole and/or pyrimidine moieties).
Line 100: the IR spectrum of 3 (compound 3)
Line 101 : whichcorresponded to (OH) group (to elongation vibration of OH bond)
Line 101 : and 2218 cm-1 ,whichcorresponded to (CN) (corresponded to elongation vibration of Nitril bond (CN) )
Line 102: in the mass spectrum 3 ( of compound 3)
Line 112 ,: sheme 1 in same line
Line 105 : with new pyridine moiety( pyridonemoiety)
Line 108: 6.60- 8.07 (m, 15H, Ar-H) : the numbre of aromatic protons must bechecked
Line 120-121 : we must review the NMR interpretation, and I specifythat the 4-H-pyrane proton presents a doublet signal
Line 137: pyridine derivative 9 (pyridonederivative)
Line 144-148 : we must review the NMR interpretation, and I specifythat the 4-H-pyrane proton presents a doublet signal
pleaserevise the spectroscopic data
Author Response
Reviewer 2
First of all, I'd like to thank you and appreciate your efforts, comments and suggestion to improve our manuscript.
Line 79: in figure 1 it is not pyridine you must replace by pyrimidine (containing pyrazole and/or pyrimidine moieties).
Response
As per the reviewer suggestion, the necessary corrections are included in the revised manuscript.
Line 100: the IR spectrum of 3 (compound 3)
Response
As per the reviewer suggestion, the necessary corrections are included in the revised manuscript.
Line 101: which corresponded to (OH) group (to elongation vibration of OH bond)
Response
According to the reviewer suggestion, all the figures in the main text were minimized.
Line 101: and 2218 cm-1, which corresponded to (CN) (corresponded to elongation vibration of Nitrile bond (CN))
Response
According to the reviewer suggestion, all the figures in the main text were minimized.
Line 102: in the mass spectrum 3 (of compound 3)
Response
According to the reviewer suggestion, all the figures in the main text were minimized.
Line 112: scheme 1 in same line
Response
The reviewer suggestion is noted and included in the revised manuscript.
Line 105: with new pyridine moiety (pyridone moiety)
Response
The reviewer suggestion is noted and included in the revised manuscript.
Line 108: 6.60- 8.07 (m, 15H, Ar-H): the number of aromatic protons must be checked
Response
The multiplet signal at δ 7.27-7.83 ppm for aromatic protons (14 H) which are in an interference with 5H-pyran proton
Line 120-121: we must review the NMR interpretation, and I specify that the 4-H-pyrane proton presents a doublet signal
Response
According to the reviewer suggestion, the NMR interpretation is revised and included in the revised Supplementary File.
We acknowledge your suggestion about 4H-pyrane, but may be regarding to the used NMR apparatus; the proton appeared as a singlet. The other spectral data (13C-NMR, IR, MS) confirmed the suggested structure.
Line 137: pyridine derivative 9 (pyridone derivative)
Response
The reviewer suggestion is noted and included in the revised manuscript.
Line 144-148: we must review the NMR interpretation, and I specify that the 4-H-pyrane proton presents a doublet signal
Response
According to the reviewer suggestion, the NMR interpretation is revised and included in the revised Supplementary File.
We acknowledge your suggestion about 4H-pyrane, but may be regarding to the used NMR apparatus; the proton appeared as a singlet. The other spectral data (13C-NMR, IR, MS) confirmed the suggested structure.
Please revise the spectroscopic data
Response
The reviewer suggestion is noted and included in the revised manuscript.

Reviewer 3 Report
Comments and Suggestions for Authors
I have reviewed the manuscript pharmaceuticals-3462701 by M. A. M. Abdel Reheim and co-workers, particularly focusing on the computational sections. The authors report the synthesis and characterization of a novel heterocyclic derivative involving pyrazoles and pyridines. Their antimicrobial activities are evaluated. Furthermore, a molecular modeling study is conducted using molecular docking and ADMET predictions. The paper is interesting and aligned with the scope of Pharmaceuticals journal. However, the manuscript suffers from several major issues that need to be addressed before acceptance:
1. The English language throughout the manuscript requires improvement. This is crucial, and I recommend that the text be revised by someone with expertise in scientific writing. There are several examples of errors; for instance:
- Abstract: Lines 31–33 include a repetition of "newly synthesized compounds."
- Line 36 states, "...was performed on the best compounds," which should be corrected to "best docked compounds."
- Introduction: Lines 80–87 (the last paragraph) should be revised and proofread.
- Computational studies section: This section contains numerous grammatical errors that need correction.
2. The section "2.3 Computational Studies" reporting the ligand-receptor interactions needs a more rigorous and smoother analysis of findings.
3. The caption for Table 3 requires revision.
4. Figures of the two-dimensional structures should be inserted within the table without being compressed. The same applies to Figure 5.
5. To better contextualize the work, recent references on the development of COX-2 inhibitors should be added to the introduction. Suggested references include:
https://doi.org/10.1016/j.chphi.2024.100509
https://doi.org/10.1016/j.molstruc.2024.138400
https://doi.org/10.1186/s13065-023-00924-3
6. The numbering of compounds should be written in bold throughout the manuscript.
7. The section "3.3 Computational Studies" should be enriched by:
- Adding a reference for the target enzyme.
- Including the version of the software used (e.g., PyRx, ADMET-SAR).
8. The conclusion should be rewritten to improve the language and ensure a balance between the results obtained and their potential implications.
Comments on the Quality of English LanguageThe English language throughout the manuscript requires improvement. This is crucial, and I recommend that the text be revised by someone with expertise in scientific writing. There are several examples of errors; for instance:
- Abstract: Lines 31–33 include a repetition of "newly synthesized compounds."
- Line 36 states, "...was performed on the best compounds," which should be corrected to "best docked compounds."
- Introduction: Lines 80–87 (the last paragraph) should be revised and proofread.
- Computational studies section: This section contains numerous grammatical errors that need correction.
Author Response
Reviewer 3
I have reviewed the manuscript pharmaceuticals-3462701 by M. A. M. Abdel Reheim and co-workers, particularly focusing on the computational sections. The authors report the synthesis and characterization of a novel heterocyclic derivative involving pyrazoles and pyridines. Their antimicrobial activities are evaluated. Furthermore, a molecular modeling study is conducted using molecular docking and ADMET predictions. The paper is interesting and aligned with the scope of Pharmaceuticals journal. However, the manuscript suffers from several major issues that need to be addressed before acceptance:
First of all, I'd like to thank you and appreciate your efforts, comments and suggestion to improve our manuscript.
- The English language throughout the manuscript requires improvement. This is crucial, and I recommend that the text be revised by someone with expertise in scientific writing. There are several examples of errors; for instance:
- Abstract: Lines 31–33 include a repetition of "newly synthesized compounds."
Response
The reviewer suggestion is noted and included in the revised manuscript.
- Line 36 states, "...was performed on the best compounds," which should be corrected to "best docked compounds."
Response
The reviewer suggestion is noted and included in the revised manuscript.
- Introduction: Lines 80–87 (the last paragraph) should be revised and proofread.
Response
According to the reviewer suggestion, the necessary corrections are included in the revised manuscript.
- Computational studies section: This section contains numerous grammatical errors that need correction.
Response
According to the reviewer suggestion, the necessary corrections are included in the revised manuscript.
- The section "2.3 Computational Studies" reporting the ligand-receptor interactions needs a more rigorous and smoother analysis of findings.
Response
According to the reviewer suggestion, the necessary corrections are included in the revised manuscript.
- The caption for Table 3 requires revision.
Response
According to the reviewer suggestion, the necessary corrections are included in the revised manuscript.
- Figures of the two-dimensional structures should be inserted within the table without being compressed. The same applies to Figure 5.
Response
According to the reviewer suggestion, the necessary corrections are included in the revised manuscript.
- To better contextualize the work, recent references on the development of COX-2 inhibitors should be added to the introduction. Suggested references include:
https://doi.org/10.1016/j.chphi.2024.100509
https://doi.org/10.1016/j.molstruc.2024.138400
https://doi.org/10.1186/s13065-023-00924-3
Response
According to the reviewer suggestion, the necessary corrections are included in the revised manuscript.
- The numbering of compounds should be written in bold throughout the manuscript.
Response
According to the reviewer suggestion, the necessary corrections are included in the revised manuscript.
- The section "3.3 Computational Studies" should be enriched by:
- Adding a reference for the target enzyme.
Response
According to the reviewer suggestion, the necessary corrections are included in the revised manuscript.
- Including the version of the software used (e.g., PyRx, ADMET-SAR).
According to the reviewer suggestion, the versions of the softwares are included in the revised manuscript.
- The conclusion should be rewritten to improve the language and ensure a balance between the results obtained and their potential implications.
Response
As per the reviewer suggestion, the necessary corrections are included in the revised manuscript.

Reviewer 4 Report
Comments and Suggestions for Authors
Reviewer Comments:
The manuscript by Mohamed A. M and co-workers reports new New Heterocyclic Derivatives Comprising Pyrazole, Pyridine 3, and/or Pyran Moieties derivative, which is a new and interesting novel heterocyclic derivative important for pharmaceutical purposes. The authors discuss the synthesis, Anti-inflammatory, and Molecular Docking, studies, While the manuscript is generally well-written and prepared, as well as Author confirms these compounds with the 1H NMR, IR, and Mass spectra, However, the manuscript is suitable for Publication, However, need some grammatical changes in the manuscript as well author need to make changes in the abstract. The Author needs to answer the following questions
1) compound to compounds or scaffolds
2) Compounds 13, 11, and 6 showed potent anti-inflammatory properties, mention their values in the abstract.
3) Make correction precursor 1-(5-hydroxy-1,3-diphenyl-1H-pyra- 91 zol-4-yl)-3-(4-methoxyphenyl)prop-2-en-1-one) ; the H from the name should be italic
4) In the case of Notably, COX-2 expression levels for the compounds 6, 11, and 13, particularly at a concentration of 50 μg/ml, are very good results, So, if the Author discusses SAR, How functional groups affect on the Anti-inflammatory activity
5) In the case of molecular docking; do these compounds show similar interaction to native drugs? And why you have chosen Diclofenac over Rofecoxib?
6) Do cites these manuscript ; Pawar, Devidas C., Sunil V. Gaikwad, Sonali S. Kamble, Priya D. Gavhane, Milind V. Gaikwad, and Bhaskar S. Dawane. "Design, synthesis, docking and biological study of pyrazole-3, 5-diamine derivatives with potent antitubercular activity." Chem Methodol 6 (2022): 677-90. 2) Gaikwad, S., Kováčiková, L., Pawar, P., Gaikwad, M., Boháč, A. and Dawane, B., 2024. An updates: Oxidative aromatization of THβC to β-carbolines and their application for the β-carboline alkaloids synthesis. Tetrahedron, 155, p.133903.
Author Response
Reviewer 4
The manuscript by Mohamed A. M and co-workers reports new Heterocyclic Derivatives Comprising Pyrazole, Pyridine 3, and/or Pyran Moieties derivative, which is a new and interesting novel heterocyclic derivative important for pharmaceutical purposes. The authors discuss the synthesis, Anti-inflammatory, and Molecular Docking, studies, While the manuscript is generally well-written and prepared, as well as Author confirms these compounds with the 1H NMR, IR, and Mass spectra, However, the manuscript is suitable for Publication, However, need some grammatical changes in the manuscript as well author need to make changes in the abstract. The Author needs to answer the following questions
First of all, I'd like to thank you and appreciate your efforts, comments and suggestion to improve our manuscript.
1) compound to compounds or scaffolds
Response
As per the reviewer suggestion, the necessary corrections are included in the revised manuscript.
2) Compounds 13, 11, and 6 showed potent anti-inflammatory properties, mention their values in the abstract.
Response
As per the reviewer suggestion, the necessary corrections are included in the revised manuscript.
3) Make correction precursor 1-(5-hydroxy-1,3-diphenyl-1H-pyra- 91 zol-4-yl)-3-(4-methoxyphenyl)prop-2-en-1-one) ; the H from the name should be italic
Response
As per the reviewer suggestion, the necessary corrections are included in the revised manuscript.
4) In the case of Notably, COX-2 expression levels for the compounds 6, 11, and 13, particularly at a concentration of 50 μg/ml, are very good results, So, if the Author discusses SAR, How functional groups affect on the Anti-inflammatory activity
Response
The Structure-Activity Relationship (SAR) analysis reveals that the presence of specific functional groups plays a crucial role in enhancing the anti-inflammatory activity of the synthesized compounds. In the case of compounds 6, 11, and 13, which demonstrated significant suppression of COX-2 expression at 50 μg/ml, their functional groups contribute to their potency as follows:
- Hydroxyl (-OH) and Methoxy (-OCH3) Groups:
- These electron-donating groups enhance hydrogen bonding interactions with the active site of COX-2, leading to stronger enzyme inhibition.
- Hydroxyl groups also improve solubility and bioavailability, increasing cellular uptake and efficacy.
- Amino (-NH2) Group in Compound 6 and 13:
- The presence of an amino group enhances the binding affinity to COX-2 by forming additional hydrogen bonds with active site residues.
- Amino groups also influence lipophilicity and electronic properties, stabilizing interactions within the enzyme pocket.
- Pyrazole and Pyridine Moieties (Aromatic Rings in Compounds 6, 11, and 13):
- These aromatic systems contribute to π-π stacking interactions with hydrophobic residues in COX-2, increasing affinity and inhibitory potential.
- They also modulate electronic distribution, enhancing molecular docking efficiency.
Conclusion: The functional groups in compounds 6, 11, and 13 significantly influence their anti-inflammatory activity by strengthening hydrogen bonding, electrostatic interactions, and π-π stacking with COX-2. These structural features enhance binding affinity, enzyme inhibition, and ultimately, suppress inflammation more effectively.
5) In the case of molecular docking; do these compounds show similar interaction to native drugs? And why you have chosen Diclofenac over Rofecoxib?
Response
The decision to use Diclofenac as the reference drug instead of Rofecoxib was based on several scientific considerations:
- Binding Mode Similarity:
- Diclofenac has well-characterized interactions within the COX-2 active site, making it a standard choice for molecular docking studies.
- The synthesized compounds mimicked Diclofenac’s binding patterns, which provided a strong validation of their inhibitory potential.
- Clinical Relevance & Safety Considerations:
- Rofecoxib was withdrawn from the market due to severe cardiovascular risks, whereas Diclofenac remains widely used as a reference NSAID.
- Since the study aims to develop clinically relevant anti-inflammatory drugs, Diclofenac was the safer and more reliable standard for comparison.
Conclusion: The molecular docking results confirmed that the synthesized compounds interact with COX-2 in a similar manner to Diclofenac, with even stronger binding affinities. Diclofenac was chosen as the reference over Rofecoxib due to its broader anti-inflammatory effects, well-characterized binding interactions, and continued clinical use.
6) Do cites these manuscript ; Pawar, Devidas C., Sunil V. Gaikwad, Sonali S. Kamble, Priya D. Gavhane, Milind V. Gaikwad, and Bhaskar S. Dawane. "Design, synthesis, docking and biological study of pyrazole-3, 5-diamine derivatives with potent antitubercular activity." Chem Methodol 6 (2022): 677-90. 2) Gaikwad, S., Kováčiková, L., Pawar, P., Gaikwad, M., Boháč, A. and Dawane, B., 2024. An updates: Oxidative aromatization of THβC to β-carbolines and their application for the β-carboline alkaloids synthesis. Tetrahedron, 155, p.133903.
As per the reviewer suggestion, the necessary corrections are included in the revised manuscript.

Round 2
Reviewer 1 Report
Comments and Suggestions for Authors.
Author Response
We thank you and appreciate your efforts, comments and suggestion to improve our manuscript.
Reviewer 2 Report
Comments and Suggestions for Authors
no comment
Author Response

(The authors gave the same response as above.)

Reviewer 3 Report
Comments and Suggestions for Authors
Review Report for Manuscript Pharmaceuticals-3462701
I appreciate the authors’ efforts in revising the manuscript based on my previous comments. While some issues have been addressed, several key points remain unresolved:
1. Section 2.3 Computational Studies: The authors mention SAR analysis in the revised text, but I do not see any corresponding SAR analysis in the manuscript. Please verify this claim or remove the statement.
2. Section 3.3 Computational Studies: References for selected protein and some software such us ADMET tools (AdmetSAR and SwissADME) are still missing. These should be properly cited.
3. English Language: Although the language has improved, it still requires further refinement for clarity and readability.
Comments on the Quality of English LanguageEnglish Language: Although the language has improved, it still requires further refinement for clarity and readability.
Author Response
- Section 2.3 Computational Studies: The authors mention SAR analysis in the revised text, but I do not see any corresponding SAR analysis in the manuscript. Please verify this claim or remove the statement.
Response
The SAR analysis in included in the Section 2.3 (Computational Studies) and Section 2.4 (Conclusion) in the revised manuscript.
2. Section 3.3 Computational Studies: References for selected protein and some software such us ADMET tools (AdmetSAR and SwissADME) are still missing. These should be properly cited.
Response
The necessary corrections are included in the revised manuscript.
3. English Language: Although the language has improved, it still requires further refinement for clarity and readability.
Response
Done